# Preparation and Laboratory Testing of Polymeric Scale Inhibitor Colloidal Materials for Oilfield Mineral Scale Control

**DOI:** 10.3390/polym14194240

**Published:** 2022-10-10

**Authors:** Hanji Wang, Huaxia Dong, Xianbin Liu, Ping Zhang

**Affiliations:** 1School of Marine and Environmental Sciences, Tianjin University of Science and Technology (TUST), Tianjin 300457, China; 2Key Laboratory of Marine Resource Chemistry and Food Technology, Tianjin University of Science and Technology (TUST), Tianjin 300457, China; 3Department of Civil and Environmental Engineering, Faculty of Science and Technology, University of Macau, Macau, China

**Keywords:** mineral scale, scale inhibitor, polymer, colloid, transport

## Abstract

Mineral scale refers to the hard crystalline inorganic solid deposit from the water phase. Although scale formation is very common in the natural environment, deposited scale particles can seriously threaten the integrity and safety of various industries, particularly oilfield productions. Scale deposition is one of the three most serious water-related production chemistry threats in the petroleum industry. The most commonly adopted engineering approach to control the scale threat is chemical inhibition by applying scale inhibitor chemicals. Aminophosphonates and polymeric inhibitors are the two major groups of scale inhibitors. To address the drawbacks of conventional inhibitors, scale inhibitor colloidal materials have been prepared as an alternative delivery vehicle of inhibitors for scale control. Quite a few studies have reported on the laboratory synthesis and testing of scale inhibitor colloidal materials composed mainly of pre-precipitated metal-aminophosphonate solids. However, limited research has been conducted on the preparation of polymeric inhibitor-based colloidal materials. This study reports the synthesis approach and laboratory testing of novel polystyrene sulfonate (PSS) based inhibitor colloidal material. PSS was selected in this study due to its high thermal stability and calcium tolerance with no phosphorus in its molecule. Both precipitation and surfactant surface modification methods were employed to prepare a barium-PSS colloidal inhibitor (BaPCI) material with an average diameter of several hundred nanometers. Experimental results indicate that the prepared BaPCI material has a decent migration capacity in the formation medium, and this material is superior to the conventional PSS inhibitor in terms of inhibitor return performance. The prepared novel BaPCI material has a great potential to be adopted for field scale control where environmentally friendly, thermal stable, and/or calcium tolerating requirements should be satisfied. This study further expands and promotes our capacity to fabricate and utilize functional colloidal materials for mineral scale control.

## 1. Introduction

Mineral scale refers to the hard crystalline inorganic solid deposit from the water phase [1,2]. Scale can form on any surface with which aqueous solution can be in contact. Although scale formation is a ubiquitous phenomenon in the natural environment, deposited scale particles can lead to severe threats to the safety and integrity of various industries, particularly oilfield productions [3,4]. In fact, scale deposition is one of the three most serious water-related production chemistry threats in the petroleum industry [3]. Scale formation and the subsequent deposition can lead to the reduction of pipe throughput and impairment of equipment functionality. In oilfield operations, the scale can deposit on the inner surfaces of production tubing and pipes, leading to a significant production loss [5]. Moreover, the scale can occur in the formation of pore space, resulting in serious damage to the reservoir formation [6]. As a result, billions of dollars were lost each year due to scale-related deferred productions and operating expenses in the petroleum industry alone [7]. Some of the most commonly observed mineral scales in the field include calcium carbonate (calcite), calcium sulfate, barium sulfate (barite), and calcium phosphate [2,8]. The most effective approach to combat the scale threats is chemical inhibition by applying scale inhibitor chemicals, which are a group of specialty chemicals capable of delaying scale formation and deposition kinetics [9,10,11]. Different from chelating chemicals, scale inhibitors belong to the category of threshold inhibitors in that scale inhibitors can be effective at a concentration of only a few milligrams per liter or below [2]. There are generally two major groups of scale inhibitors, including aminophosphonate and polymeric inhibitors [8]. It is commonly believed that aminophosphonate inhibitors are good crystal growth inhibitors, while polymeric inhibitors are good nucleation inhibitors and scale dispersants [11,12]. To apply scale inhibitors into the production system for scale control, one common approach of inhibitor delivery is to continuously inject inhibitors propelled by a chemical pump [12]. Although continuous injection is straightforward to implement with a low operational cost, this method cannot deliver inhibitor chemicals into downhole formation for reservoir scale control [3,9]. In oilfield operations, in order to place inhibitor into formation, a scale squeeze treatment is normally conducted by pumping (squeezing) inhibitor into the formation through the perforations at the bottom of the production tubing [10,11,12]. In a typical squeeze treatment, the production well will be shut down, and a volume of solution will be pumped through the production tubing into the formation medium to condition the formation (also called preflush treatment). The injected preflush solution can push the production fluid back into the reservoir and clean up the surfaces of the formation [13]. Subsequent to preflush treatment, a volume of chemical inhibitor will be pumped into the formation. Next, another volume of brine solution will be injected into the formation to push the injected chemical inhibitor away from the tubing into a deeper reservoir (also called overflush treatment). Upon completion of overflush treatment, the well will be shut-in for a period of time to allow the attachment of the delivered inhibitor onto the surface of the formation medium. Afterward, the well production will be resumed with the production fluid flowing back into the well (also called flow back) [11,12]. During the flow of production fluid from the reservoir into the well, the delivered inhibitor will be gradually released from the pore space into the production fluid and flow into the well for scale control (also called inhibitor return) [14]. With the progression of inhibitor return, the aqueous concentration of inhibitor in the production fluid (return concentration) will progressively decline over time. When the return concentration is low enough, another squeeze treatment might be necessitated to elevate the inhibitor return concentration to effectively control scale threats [15]. The time duration between two consecutive squeeze treatments is called the squeeze lifetime. Obviously, a longer squeeze lifetime is of great technical and financial advantage in oilfield scale control [9,14].

Although squeeze treatment has been the standard operation in the oilfield to control scale threats downhole and in the well, there are a number of drawbacks associated with conventional squeeze treatment. One of the most pronounced drawbacks is that often more than one-third of the delivered inhibitor will flow out of the formation, together with the production fluid, during the very early stage of inhibitor return due to a poor attachment of inhibitor on the surfaces of formation materials, leading to a considerable waste of inhibitor and hence a reduced squeeze lifetime [9,14,16]. To address the shortcomings of conventional squeeze treatment, scale inhibitor colloidal materials have been prepared as an alternative to the conventional chemical inhibitor to be deployed during squeeze treatment. Most of the reported scale inhibitor colloidal materials are colloidal particles composed of pre-precipitated cation-inhibitor with submicron particle sizes [17]. A number of authors have reported different synthesis routes to prepare inhibitor colloidal or nano- materials with different compositions [17,18,19,20,21]. Two key characteristics of these colloidal inhibitor materials are important to potential field applications. One is the transportability of these colloidal materials in the formation of porous media. Colloidal materials serve as a delivery vehicle of the embedded chemical inhibitor, and a desirable migration capacity of these materials is substantially beneficial in placing inhibitors into the target zone in formation for scale control [18]. The other key property is the return behavior of the colloidal materials to release the inhibitor into the production fluid in a gradual and controlled manner with an extended squeeze lifetime [21]. These colloidal inhibitor materials typically have a relatively stable aqueous solubility in the production fluid, allowing these materials to gradually release chemical inhibitors into the production fluid controlled by material aqueous solubility. Hence, compared with conventional chemical inhibitors, inhibitor colloidal materials can have a substantially prolonged squeeze lifetime [17]. 

As an important group of scale inhibitors, most polymeric inhibitors contain little or no phosphorus element, making these materials substantially more environmentally friendly compared with aminophosphonates. Thus many polymeric inhibitors are regarded as “green inhibitors” [4,5,22,23]. Moreover, due to the versatile molecular structures of polymeric inhibitors, many polymeric inhibitors have unique advantages over aminophosphonates, such as a higher calcium tolerance, higher thermal stability, and a wider range of pH tolerance [8]. According to a literature study, most of the existing inhibitor colloidal materials are prepared via precipitation of cations with aminophosphonate inhibitors, with limited research on preparing polymeric inhibitor-based colloidal materials [17,18,19,20,21]. To fill this gap, in this study, we studied the synthesis approach and the laboratory testing of novel polystyrene sulfonate (PSS) based colloidal inhibitor material. PSS is a polymeric inhibitor commonly adopted for industrial scale control. PSS was selected in this study due to its high thermal stability and calcium tolerance with no phosphorus in its molecule [8]. Both precipitation and surfactant surface modification methods were employed to prepare a barium-PSS colloidal inhibitor (BaPCI) material. The optimal synthesis conditions for BaPCI have been carefully investigated. Furthermore, both transportability and inhibitor return behaviors of the synthesized BaPCI material in the formation porous medium have been systematically evaluated under representative oilfield operating conditions. Experimental results indicate that the prepared BaPCI material has a decent migration capacity in the formation medium, and this material is superior to the conventional PSS inhibitor in terms of squeeze lifetime. To the best of our knowledge, this is the first report on synthesizing and testing metal-polymeric inhibitor colloidal material for scale control. The prepared novel BaPCI material has a great potential to be adopted for field scale control where environmentally friendly, thermal stable, and/or calcium tolerating requirements should be satisfied. This study further expands and promotes our capacity to fabricate and utilize functional colloidal materials for mineral scale control. 

## 2. Materials and Methods

### 2.1. Chemicals

Polystyrene sulfonate (PSS) with 30% activity (wt./wt.) and a molecular mass of 35,000 Da were purchased from Macklin Biochemical Co., Ltd. (Shanghai, China) and used as a scale inhibitor. The molecular structure of PSS is shown as Appendix A. Sodium dodecyl benzenesulfonate (SDBS) (>95%) was purchased from Meryer Chemical Technology Co., Ltd. (Shanghai, China). Chemicals such as barium chloride, calcium chloride, sodium chloride, nitric acid, hydrochloric acid, sodium hydroxide, sodium sulfate, and potassium bromide were in analytical grade and purchased from Macklin Biochemical Co., Ltd. (Shanghai, China). Calcite solid with a purity of 98.5% was purchased from Yanxi Mining Co., Ltd. (Shijiazhuang, China). Deionized water (DI water) was obtained using a Millipore Direct-Q 3UV system (Merck, Billerica, MA, USA).

### 2.2. Preparation of BaPCI and Its Suspension

In a typical BaPCI synthesis, 4.6 mL of PSS solution (30% wt.) was added dropwise into 50 mL of BaCl_2_ solution (0.3 M) under constant stirring using a magnetic stir bar in a glass beaker. By mixing PSS solution with BaCl_2_ solution, Ba-PSS white precipitate started to occur. Upon the completion of the PSS solution addition, we continued to mix the mixture solution for 30 min. Subsequently, the mixture solution was aged at 80 °C condition for 12 h. Next, the formed precipitate was collected by centrifugation at 5000 rpm, followed by DI water washing for 3 times to remove the unreacted PSS or BaCl_2_ from the solid surface. The resulting wet slurry was then air dried at 80 °C overnight, and the obtained dry powder was BaPCI. Finally, to obtain the aqueous suspension of BaPCI, a certain amount of BaPCI dry powder was dispersed into an aqueous solution containing SDBS surfactant (0.2% wt.) via ultrasonication treatment (Q800, Qsonica, Newtown, CT, USA). This mixture solution was sonicated at 4 °C for 10 min, and a BaPCI suspension was obtained. The impacts of SDBS surfactant concentration, sonication duration, and presence of KCl were evaluated for their roles in influencing the physiochemical properties of BaPCI and its suspension. The BaPCI suspension adopted for the column transport test and laboratory squeeze simulation experiment was prepared by dispersing the obtained BaPCI into an aqueous solution containing 0.2% SDBS and 0.2% KCl via sonication treatment for 10 min. Note that the BaPCI aqueous concentration in this study was reported as the aqueous concentration of PSS embedded in the BaPCI material in the unit of mg L^−1^.

### 2.3. BaPCI Scale Inhibition Performance and Inhibition Efficacy Tests

The scale inhibition performance of the prepared BaPCI in terms of inhibiting calcium phosphate or calcite scale was evaluated and compared against that of the PSS inhibitor. The experimental procedures follow what was previously reported [24]. The detailed experimental procedures can be found in Appendix A. Section S3 also elaborates on the determination of scale inhibition performance efficiency values (η, %) for both calcite and calcium phosphate scale tests. Moreover, to further evaluate the inhibition efficacy of the prepared BaPCI material, a number of tests were carried out with different aqueous concentrations of BaPCI to inhibit the barite scale. A series of barite nucleation (turbidity) tests have been conducted to measure the induction time with different aqueous concentrations of BaPCI or PSS at an ambient condition. Induction time (t_ind_, min) is defined as the time interval between the establishment of supersaturation (i.e., mixing Ba species with SO_4_ species) and the onset of the nucleation (i.e., detection of the formed scale particles by instruments or visually [25]. The turbidity tests in this study followed the procedure previously reported involving a turbidity meter (TL2300, HACH Co., Loveland, CO, USA) [26,27]. Briefly, 2.5 mL BaCl_2_ solution (0.6 mM) was mixed with 2.5 mL Na_2_SO_4_ solution (0.6 mM) in the presence of either BaPCI or PSS inhibitor at room temperature. The turbidity of the mixture solution was continuously measured for up to 1 h. The detailed experimental procedure can be found in Appendix A.

### 2.4. BaPCI Column Transport Experiments

The migration performance of the prepared BaPCI was examined via laboratory transport experiments in a calcite-packed column. The column setup includes a syringe, a syringe pump (Guanjie Co., Zibo, China), a glass column (Xiamei Co., Shanghai, China), and a fraction collector (DBS-160, Jiapeng Technology Inc., Shanghai, China), similar to our previous studies [18,21]. The column setup schematic is shown as Appendix A. Calcite solid was ground into fine particles and chosen as the column packing material. The grain size distributions of the calcite particles were 100–150 μm and 50–100 μm. The calcite particles were washed with acetic acid (1% wt.) followed by DI water rinsing three times to remove fine grains before being dried in an oven at 80 °C for 24 h. Next, the cleaned calcite particles with a mass of ca. 10.167 g were packed into the glass column with 1 cm in inner diameter and 7.5 cm in length. Two stainless steel frits with a pore size of 10 μm were placed at both ends of the column to maintain calcite particles inside the column. Initially, a non-reactive tracer potassium bromide (KBr) test was conducted, and the characteristics of the packed calcite column, such as pore volume (PV, mL) and dispersion coefficient (D, cm^2^ min^−1^), were determined via the KBr tracer test. PV is the empty volume of the column unoccupied by calcite particles and can be calculated as the total column volume subtracted by the volume occupied by calcite particles. Prior to the BaPCI column transport experiments, the glass column packed with calcite particles was flushed with a KCl solution (0.2%) for 3 PVs for column conditioning (the preflush treatment). Subsequently, BaPCI transport experiments were conducted by injecting the prepared BaPCI suspension into the calcite-packed column for up to 5 PVs. The fraction collector collected the effluent solution to determine the effluent BaPCI concentration to establish a BaPCI breakthrough curve. This study studied the impact of flow rates by pumping the BaPCI suspension at different flow rates ranging from 6.5 mL h^−1^ to 113 mL h^−1^, corresponding to a linear flow velocity of 5.5 m d^−1^ to 96 m d^−1^. The temperature conditions evaluated include 4, 20, 50, and 70 °C. 

### 2.5. Laboratory Squeeze Simulation Experiments of BaPCI

The ability of the prepared BaPCI to return the PSS inhibitor into the production fluid for scale control was evaluated via a set of laboratory squeeze simulation experiments (LSSEs). A totally contained squeeze simulation approach was adopted in a calcite medium, similar to the previous studies [17,21]. The apparatus consists of a glass column (1 cm by 7.5 cm), a pneumatic pump (DIPump550, Kamoer Fluid Tech, Shanghai, China), a water bath, and a fraction collector. To begin with, about 10.167 g of ground calcite particles with a grain size ranging from 100 to 150 μm were packed into the glass column with a height of 7.5 cm. Two stainless steel frits with a pore size of 10 μm were placed at both ends of the column. This packed column was subsequently pre-saturated with a synthetic brine composed of 1 M NaCl, 0.1 M CaCl_2_, 25 mM KCl, and 0.1 M NaHCO_3_ with an initial solution pH of 5.5 at 70 °C condition. Subsequent to synthetic brine pumping for 3 PVs, 0.5 PV of the prepared BaPCI suspension with 1.1% (wt.) PSS was injected into the packed calcite column at 70 °C condition to initiate LSSE. Immediately after the BaPCI suspension injection, another 0.5 PV of NaCl solution (1 M) was injected into the column to push the injected BaPCI suspension to the other half of the column (the overflush treatment). Subsequently, the column was shut-in at 70 °C for 24 h to allow the injected BaPCI to affix to the surface of calcite particles inside the column. Afterward, the column was flushed with the synthetic brine from a reverse direction at 70 °C to simulate the production fluid flowing over the formation materials (flow back) post a squeeze treatment in the field. During the synthetic brine flushing, the brine flow rate was 30 mL h^−1^, corresponding to a linear flow velocity of 25.5 m d^−1^. The effluent solution was collected by the fraction collector and analyzed for PSS concentration so that a BaPCI inhibitor return curve could be established. The schematic of the squeeze simulation procedure is shown as Appendix A. As a comparison, a similar LSSE was performed by adopting a PSS inhibitor solution with 0.97% (wt.) PSS instead of BaPCI suspension. 

### 2.6. Analytical and Characterization Methods

Concentrations of barium and sulfur in the samples were measured by inductively coupled plasma-optical emission spectrometer (PRODIGY-H, Leeman, Hudson, NH, USA). The aqueous concentration of the PSS inhibitor was calculated based on the measured sulfur concentration. The particle size and zeta potential of the prepared BaPCI particles were determined by dynamic light scattering and Zeta-PALS (90Plus Zeta, Brookhaven Instruments Corporation, Holtsville, NY, USA) with a measurement uncertainty of less than 5%. Transition electron microscopy (TEM) was performed on an electron microscope (F30, FEI Co., Hillsboro, OR, USA) at 130 kV. Samples for TEM analysis were prepared by dripping the BaPCI suspension onto a copper grid coated with amorphous carbon-holey film. Scanning electron microscopy (SEM) (Quanta FEG 250, FEI Co., Hillsboro, OR, USA) was used to study the morphology of the obtained solids. Fourier transform infrared (FT-IR) spectra were investigated on an FT-IR spectrometer (VERTEX 70v, Bruker Corp., Billerica, MA, USA) with the KBr pellet technique with a spectrometer range from 4000 to 400 cm^−1^. Thermogravimetric analysis (TGA) and differential scanning calorimetry (DSC) analyses were carried out using a thermal analysis instrument (STA-449-F5, NETZSCH, Selb, Germany). Samples were heated with a heating rate of 10 °C min^−1^ from 35 °C to 900 °C in an atmosphere of flowing argon gas (100 mL min^−1^). X-ray photoelectron spectroscopy (XPS) analysis (ESCALAB 250Xi K-alpha, Thermo Fisher Scientific, Waltham, MA, USA) was undertaken under a high vacuum on graphene using Al Kα 1486.6 eV radiation at 400 W (15 kV).

## 3. Results and Discussion

### 3.1. Optimization of Synthetic Conditions for BaPCI and Its Suspension

In this study, polymeric scale inhibitor PSS was adopted to prepare Ba-PSS colloidal inhibitor material via precipitation coupled with surface modification by ultrasonication in the presence of SDBS surfactant, as depicted in Figure 1. The resultant BaPCI suspension is a stable white suspension with a solution pH of 7.7. Based upon the elemental analysis, the molar ratio of Ba to PSS monomer of the BaPCI material is determined to be 1:2. In view of the PSS molecular structure, two possible structures of Ba-PSS precipitate via intermolecular and intramolecular attractions were illustrated in Schematic 1a. Efforts have been made to investigate the optimal BaPCI synthesis condition in terms of SDBS amount, presence of KCl, and sonication duration. Figure 2a presents the impact of SDBS surfactant amount on the properties of the prepared BaPCI in terms of particle size and zeta potential. It is well known that a reduction in particle size and the zeta potential for colloidal particles represents a more desirable colloidal stability [28]. Evidently, the presence of SDBS can effectively modify BaPCI surface chemistry and reduce both particle size and zeta potential, rendering BaPCI particles with enhanced stability. Both electrostatic repulsion and steric hindrance can account for enhanced stability due to SDBS presence [18]. As shown in Figure 2a, the optimal SDBS concentration occurred at 0.2% (wt.), where the particle size of 325 nm and zeta potential of −80.5 mV was the lowest within the range of SDBS concentrations studied. Thus, the SDBS concentration for BaPCI synthesis was determined to be 0.2%. Figure 2b shows the impact of KCl electrolyte on BaPCI stability in the presence of 0.2% SDBS. The presence of electrolytes in a colloidal system would suppress the electric double layer and reduce the zeta potential, leading to reduced colloidal stability [18]. This conclusion is in line with the experimental observation that adding KCl into the BaPCI suspension can considerably increase particle size and zeta potential. However, the KCl electrolyte is the necessary background electrolyte in the inhibitor package to prevent clay swelling by maintaining mineral permeability and ion exchange selectivity [3,5]. In view of the detrimental effect on colloidal stability and the necessity of including KCl in inhibitor suspension, a KCl concentration of 0.2% was selected for BaPCI suspension preparation, which would result in a particle size of 400 nm and a zeta potential of −63 mV. Furthermore, the impact of ultrasonication duration on BaPCI size and zeta potential has been evaluated. Sonication treatment has been extensively adopted in the synthesis of various colloidal and nanomaterials [29]. Sonication treatment can profoundly modify the morphology as well as the stability of colloidal materials. As illustrated in Figure 2c, in the presence of 0.2% SDBS and 0.2% KCl, a prolonged sonication time from 0 min to 10 min can substantially reduce BaPCI particle size. A sonication time between 10 min and 20 min seems to have a less significant impact on particle size. Considering the effect of sonication duration on the particle size reduction and the potential BaPCI manufacturing cost associated with sonication treatment, the sonication time in this study was chosen to be 10 min, which could effectively reduce the particle size to 230 nm. As an interim summary, the BaPCI prepared in the presence of 0.2% SDBS and 0.2% KCl with 10 min ultrasonication treatment has a particle size of 230 nm and zeta potential of −63 mV. 

### 3.2. Characterization of the Prepared BaPCI and Its Suspension

The morphologies of the prepared BaPCI material were examined via TEM and SEM characterizations. Based on the SEM microimage (Figure 3a), the prepared BaPCI material is in an approximately spherical shape with an average diameter of 200–300 nm, which is in line with the particle size analysis (Figure 2c). This suggests that the attractive interaction of Ba species with the sulfonate groups of the PSS polymer molecules can lead to the aggregation of the polymeric cluster into a spherical shape. The presence of SDBS coupled with sonication treatment can effectively regulate the particle size within 250 nm. TEM microimage (Figure 3b) suggests that the prepared BaPCI material bears a spherical core with irregularly shaped surrounding materials possibly composed of SDBS surfactant. Considering the microscopic analysis results, a schematic of BaPCI molecular composition can be illustrated in Figure 1b, where the PSS molecules bonded with Ba species constitute the inner core and the SDBS surfactant molecules cover the colloid surfaces. 

The skeleton structures of the BaPCI material can be unveiled by the FI-IR characterization. The FT-IR spectra (Figure 3c) indicate that the BaPCI materials demonstrate several bands visible at several wave numbers, such as 3432, 2921, 1601, and 833 cm^−1^. Referencing the FT-IR group frequency table [30], the corresponding assignment of these observed frequencies can be interpreted: the band at 3432 cm^−1^ is assigned to the vibration of the hydrogen bond in the PSS molecule; the band located at 2921 cm^−1^ corresponds to the asymmetric stretching vibration of ethyl group in PSS molecule; the band at 1601 cm^−1^ is due to aromatic benzene ring skeletal vibrations; the band located at 833 cm^−1^ can be attributed to the bending of the para-substituted benzene ring. The presence of these four bands suggests that BaPCI material still carries several characteristic groups of PSS monomer, such as ethyl group and benzene. A comparison of the FT-IR spectra of BaPCI and PSS shows that a noticeable shift can be observed at the wave numbers between 1200 and 1000 cm^−1^, particularly at 1176, 1039, and 1008 cm^−1^ [31]. These bands are attributed to the S-O stretching vibration of the sulfonate group [32]. The shift in wave numbers associated with the sulfonate group was due to the interaction of Ba species with the sulfonate group of PSS, forming the Ba-O-S bond as illustrated in Figure 1. XPS was employed to examine the valence state of elements in BaPCI. By examining the valence states of O, C, and S elements (i.e., O 1s, C 1s, S 2p), it was found that the most noticeable shift in elemental valence state between BaPCI and PSS could be observed for O 1s (Figure 3d). O 1s spectrum with a binding energy of 532.5 eV corresponds to the O in the PSS molecule. The shift in the O 1s spectrum from PSS (532.5 eV) to BaPCI (531.7 eV) verifies the formation of the chemical bond of Ba-O-S. Furthermore, the TGA-DSC characterization was carried out for the BaPCI material from 0 °C to 800 °C, with an approximately 55% mass loss by the end of the test (Figure 3e). The entire TGA profile can be segmented into three regions: 35 to 200 °C, 200 to 400 °C, and above 400 °C. The mass loss over the range of 0 to 200 °C could be explained by the evaporation of adsorbed water, removal of lattice water, and decomposition of organic fractions from BaPCI. It is difficult to differentiate the mass loss of each constituent due to the overlapping of these reactions within this temperature range. From 200 °C to 400 °C, the mass loss was insignificant. The mass loss started to increase rapidly at 400 °C with a large amount of heat being released according to the DSC profile, indicative of reaching the combustion point of BaPCI. The mass loss beyond 400 °C is attributed to the removal of remaining organics and sulfur elements in the material [33,34].

### 3.3. Evaluation of Scale Inhibition Performance and Efficacy of the Prepared BaPCI Material

As a delivery vehicle of the PSS inhibitor, it is vital that the prepared BaPCI possesses the functionality of controlling scale formation. In other words, the formation of Ba-O-S bound via Ba and sulfonate group interaction should not compromise the scale inhibition effect of the embedded PSS inhibitor. Thus, a set of laboratory tests have been carried out to evaluate the scale inhibition performance and efficacy of the BaPCI in terms of controlling calcium phosphate, calcite, and barite scales, which are commonly observed mineral scales in subsurface formation and oilfield squeeze operations [5]. Figure 4a compares the scale inhibition performance of BaPCI and PSS inhibitors by inhibiting calcium phosphate scale at different scaling tendencies at 80 °C, which is typical for oilfield operations. It shows that at the testing condition with 100 mg L^−1^ Ca, the curves of BaPCI and PSS mostly overlapped with each other, demonstrating a similar scale inhibition performance within the range of inhibitor concentrations evaluated. This suggests that the inhibition performance of BaPCI is on par with that of PSS, given the same inhibitor concentration. The observed similarity in the scale inhibition performance of the BaPCI and PSS inhibitors can be explained by the functioning mechanism of BaPCI materials. As elaborated above, scale colloidal inhibitor materials gradually release chemical inhibitors into the production fluid during the operations. Thus, since the onset of scale inhibition performance tests, BaPCI would gradually release PSS inhibitor into the aqueous solution. Once released from the BaPCI, the freely mobile PSS would inhibit scale formation in the same fashion and mechanism as the pristine PSS [1,14]. At elevated scaling threats with 200 and 400 mg L^−1^ Ca, the measured η values would reduce by ca. 5% and 18%, respectively, compared with that of 100 mg L^−1^ Ca in the presence of BaPCI. This observation is in line with the testing results obtained from chemical inhibitors [35]. As for the BaPCI inhibiting calcite scale test, a similar result shown in Figure 4b can be found that the inhibitory performance of BaPCI is comparable to that of PSS at 240 mg L^−1^ Ca at 80 °C. Moreover, the elevation of calcite scaling threat with a higher Ca concentration can reduce the η value. It can be seen in Figure 4b that a peak scale inhibition efficiency was observed for all the experiments of the BaPCI inhibiting calcite, especially the scenarios with 240 and 480 mg L^−1^ Ca^2+^. A plausible explanation is the formation of Ca-PSS precipitate of the Ca^2+^ species with the released PSS from BaPCI at a higher inhibitor concentration. As reported previously, scale inhibitors could readily form a precipitate with Ca^2+^ species and change the chemical behaviors of the inhibitors [21]. Once the precipitate of Ca-PSS is formed, the aqueous concentration of the freely mobile PSS is considerably reduced, leading to a lowered scale inhibition performance. Furthermore, efforts have been made to investigate BaPCI scale inhibition efficacy against the barite scale via a set of turbidity tests. A prolonged induction time (t_ind_) suggests a higher scaling control efficacy [26,35]. As indicated previously, the practical threshold for the onset of scale particle nucleation is when the reading of nephelometric turbidity units (NTUs) reaches 1 to 2 [36]. In this study, a turbidity reading of 2 NTU has been set as the onset of crystal nucleation. Figure 4c shows that the increase in the BaPCI concentration can significantly extend t_ind_ from 9 min in the control scenario (no inhibitor) to 40 min with 25 mgL^−1^ BaPCI presence. Evidently, the presence of BaPCI can effectively control barite scale nucleation, and an increase in the BaPCI concentration can lead to an extended t_ind_. These scale inhibition test results can verify that the embedded PSS inhibitor in BaPCI is still effective in controlling mineral scale threats.

### 3.4. Transport of BaPCI in Calcite Packed Column

A critical performance indicator of the prepared BaPCI material is the transport capacity of this material in subsurface formation media. As a delivery vehicle of chemical inhibitors, colloidal inhibitor materials with a desirable transport performance are capable of migrating in the formation pore space to place the chemical inhibitor into the formation target zone for scale control [17]. In this study, calcite was selected as the formation material since calcite was the most active subsurface mineral with respect to reaction with scale inhibitors [17]. In order to examine the transport of BaPCI in calcite medium, a packed bed column setup was adopted in this study, where a volume of BaPCI suspension was pumped into the column packed with calcite particles at various physiochemical conditions. Similar to previous studies investigating the flow-through of colloidal particles in a porous medium, the transport of BaPCI can be elucidated by an advection-and-diffusion mechanism [21]. Mathematically, the BaPCI transport process in the calcite-packed column can be delineated by a one-dimensional advection-dispersive equation coupled with a deposition term [37,38,39]: (1)$$R\frac{\partial C_{}}{\partial t}-D\frac{\partial^{2}C_{}}{\partial x^{2}}+v\frac{\partial C_{}}{\partial x}+J_{d}C=0$$

The physical meaning of each term of the above equation is that: C (mg L^−1^) represents the BaPCI concentration in the effluent at a given time, t (min); D (cm^2^ min^−1^) denotes the hydrodynamic dispersion coefficient; v (cm min^−1^) is the linear flow velocity; and x (cm) corresponds to the travel distance which is the length of the column of 7.5 cm. Two key parameters of critical importance in colloidal transport include R (unit less), the retardation factor, and J_d_ (min^−1^), the first-order deposition coefficient of BaPCI material to the surfaces of calcite medium. R characterizes the sorptive behavior of BaPCI to calcite surfaces due to the retardation effect [39]. J_d_ accounts for the deposition kinetics of BaPCI to calcite surfaces and is principally influenced by the energy barrier based upon the DLVO theory [38,40]. The detailed explanation and the mathematical solution of the advection-dispersive equation can be found in Appendix A. A breakthrough curve of BaPCI in calcite medium can be established by plotting the normalized effluent BaPCI concentration (i.e., C/C_0_ or breakthrough efficiency) as a function of the volume of BaPCI suspension pumped into the column (i.e., number of PVs). In this study, a systematic investigation of the BaPCI migration capacity in calcite medium was carried out by evaluating the effects of surfactant preflush, temperature, flow rate (flow velocity), and calcite grain size. Table 1 summarizes the detailed experimental conditions of each transport experiment (TE) of BaPCI in the calcite medium. For all TEs, the calcite-packed column had a dimension of 1 cm by 7.5 cm, a PV of 2.1 mL, a porosity of 36%, and the initial concentration (C_0_) of BaPCI was 200 mg L^−1^ in the influent solution. The D value is determined via the KBr tracer test (detailed in Appendix A). Table 2 presents the breakthrough efficiencies and the calculated R and J_d_ values for each TE. Figure 5a compares the transport of BaPCI and PSS inhibitor at a comparable experimental condition (TE #1 and TE #2). It shows that the BaPCI achieved a breakthrough efficiency of 7% lower than that of the PSS inhibitor at the testing condition. The observed reduced breakthrough level of BaPCI was due to a high deposition rate to the calcite surface and higher adsorption onto calcite, resulting from the retardation effect of the BaPCI material on the calcite surfaces. 

Figure 5b illustrates the BaPCI breakthrough curves in the calcite medium, emphasizing the impact of the presence of SBDS surfactant in the preflush solution (TE #2 and TE #3). Before the transport experiments, about 4 PV brine solution was pumped into the calcite-packed column to simulate the injection of preflush fluid during either a hydraulic fracturing or a squeeze treatment. In this study, the presence of SDBS (0.2% wt.) in the preflush solution in TE #3 can evidently lead to an elevation of breakthrough efficiency from 58% (TE#2) to 75%. This observed result can be further quantified by the calculated R and J_d_ values based on Equation (1). Both R and J_d_ values were determined to be lower in TE #3 (Table 2), signifying weaker sorption of BaPCI and a reduced deposition rate of BaPCI to the calcite medium surfaces. As elaborated in a previous study investigating the migration of phosphonate-based inhibitor nanomaterials in a calcite medium, it was found that preflushing the calcite medium can considerably improve the breakthrough efficiency by more than 10% at a comparable flow velocity [17]. The reason is attributed to the enhanced energy barrier of the inhibitor nanomaterials with the medium surface from the viewpoint of DLVO theory. The presence of SDBS was measured to reduce the zeta potential of calcite from −8.6 mV to −27.5 mV in a brine solution containing 33 mM (or 0.25%) KCl. The mechanism of the enhanced energy barrier in the presence of SDBS included a more prominent repulsive force and an enhanced steric hindrance [18]. It can be envisaged that, in this study, the presence of anionic surfactant SDBS in the preflush solution would coat the calcite medium surface with a layer of SDBS. In this way, an enhanced energy barrier would be expected when the BaPCI material was in contact with the calcite medium during the transport experiment in view of the fact that BaPCI was also negatively charged. In oilfield operations, preflush is a common practice prior to hydraulic fracturing or scale squeeze treatment [3]. In light of the enhanced transport efficiency with SDBS presence in the preflush solution, SDBS surfactant (0.2% wt.) was included in the preflush solution of all other TEs from #4 to #10 (Table 1).

Figure 5c illustrates the BaPCI breakthrough curves at different temperatures with otherwise similar experimental conditions (TE #3 to #6). The results suggest that the temperature plays a moderate role in influencing the BaPCI transport. The elevation in the temperature from 4 °C (TE #4) to 70 °C (TE #6) resulted in a reduction in C/C_0_ from 75.5% to 64.4%. The increase in temperature increases J_d_ values from 0.014 to 0.022, indicative of a more pronounced BaPCI deposition onto the calcite surfaces. The fact that the calculated R values did not change noticeably suggests that the elevation in temperature did not materially impact the sorptive behavior of BaPCI on the calcite surfaces. This result is similar to the previous study examining the transport of a phosphonate-based nanoparticle capsule, where a 7% reduction in transport efficiency was yielded with a temperature increase from 4 °C to 50 °C [35]. Since the intrinsic formation temperature is typically higher than the ambient temperature [3], a detrimental effect on BaPCI transportability is expected at an elevated formation temperature. Further studies are required to examine the migration capacities of the colloidal inhibitor materials at high and ultra-high temperature conditions. Figure 5d highlights the impact of flow rate (or linear flow velocity) on BaPCI transport (TE #3, #7, #8, and #9). Clearly, the flow velocity can noticeably impact the migration of BaPCI in calcite. When the flow velocity was in the range from 5.5 to 24 m d^−1^, the BaPCI breakthrough efficiencies stayed relatively stable at around 73 ± 1.3%. The corresponding R and J_d_ values increase moderately with the flow velocity increase within this range. When the flow velocity reached 96 m d^−1^ (TE #9), a more considerable reduction of C/C_0_ to 69% could be observed, with the J_d_ value being calculated as 20 times higher than that of TE #3. This suggests that at a substantially higher flow velocity, the BaPCI materials have a higher tendency to deposit onto the calcite surfaces. The conclusion from this study is different from that of a previous study investigating the transport of scale inhibitor reverse micelle in calcite medium [41]. In this previous study, the increase in the flow velocity from 10.2 m d^−1^ to 82.5 m d^−1^ led to an increase in the C/C_0_ from 37% to 60% [41]. One possible explanation for this discrepancy is that the scale inhibitor reverse micelle is a water-in-oil microemulsion with a very different physiochemical property than the BaPCI material fabricated in this study. Moreover, isooctane was adopted as the preflush fluid in the previous study. The difference in the preflush fluid could give rise to the difference in colloidal material transport in the calcite medium. Lastly, the impact of calcite grain size was evaluated at two different grain size ranges: 100–150 µm (TE #3) and 50–100 µm (TE #10) at otherwise similar experimental conditions. Figure 5e indicates that the reduction in calcite grain size will lead to a marginal reduction in the C/C_0_ from 74.6% to 68.8%, with the corresponding R values being increased from 1.20 to 1.52. This phenomenon can be attributed to the increased surface area of the calcite particles with a smaller grain size range, leading to higher adsorption of BaPCI to calcite surfaces during the transport and hence an increased R-value. 

### 3.5. Laboratory Squeeze Simulation Experiments of BaPCI

The objective of the LSSE is to evaluate the release of the scale inhibitor chemical (i.e., inhibitor return) from the prepared BaPCI material in a column setup to mimic the actual practice during field scale squeeze operations. BaPCI injection and overflush in the LSSE represent the field inhibitor injection (squeeze) to deliver the inhibitor into the formation and the subsequent overflush treatment to push the injected inhibitor into a deeper formation with a wider scale protection area [10,11,12]. The following column shut-in treatment in both the LSSE and field operation is to retain the delivered inhibitor on the surfaces of the formation materials. The flow back of the synthetic brine in LSSE is to mimic the flow of production fluid out of the reservoir formation. During this process, the production fluid would be in contact with the retained inhibitor and gradually release the retained inhibitor back into aqueous production fluid for the scale control (an inhibitor return process) [13,14,15,16]. An inhibitor return curve can be established by plotting the effluent inhibitor concentration versus the number of PVs of the brine returned during an LSSE. As detailed in Table 3, both LSSE #1 (BaPCI) and LSSE #2 (PSS solution) adopted the same column setup with the same dimension and PV and a comparable amount of PSS inhibitor (12.05 mg vs. 10.63 mg as PSS) injected into the calcite-packed column. The temperature for both LSSEs was selected as 70 °C to represent the oilfield conditions. 

Figure 6 compares the return curves associated with the two LSSEs, and Appendix A presents these return curves on a linear scale of inhibitor concentration. Appendix A plots the cumulative mass of inhibitor returned during the course of LSSEs. A considerable difference can be observed between these two LSSEs. As shown, the initial return concentration at 0.5 PV was 380 mg L^−1^, with the amount of inhibitor returned accounting for ca. 3.2% of the total inhibitor injected within the first PV of return in LSSE #1. On the contrary, the PSS return concentration was as high as 1400 mg L^−1^ at 0.5 PV, and more than 7% of the total inhibitor injected was returned within 1 PV in LSSE #2. The peak inhibitor return concentration occurred as 1550 mg L^−1^ at 2 PV and 5700 mg L^−1^ at 1 PV for LSSE #1 and #2, respectively. These experimental observations suggest that a substantially larger amount of PSS inhibitor was flushed out of the column in the early stage of inhibitor return in LSSE #2 adopting the PSS inhibitor compared with LSSE #1 employing BaPCI. As elaborated in the previous studies [21], the amount of unretained inhibitor placed at the pore space of the formation medium accounts for the inhibitor return during the early stage of LSSE or field squeeze treatment. Clearly, a much higher percentage of PSS inhibitors in LSSE #2 failed to affix to the calcite medium surface, leading to a considerably elevated initial return concentration during inhibitor flow back. Different from PSS solution, the BaPCI material demonstrates a more desirable behavior in terms of retaining to calcite surface with a reduced amount of inhibitor being flushed out of the column. Although BaPCI material shows decent transportability in calcite medium with weak adsorption to calcite surface during TEs, as illustrated in Figure 5, BaPCI can be successfully retained by calcite medium during the 24 h shut-in period. This process was designed to allow the injected inhibitor material to affix to the formation medium surface. As illustrated in Figure 6, with the further proceeding of LSSE, a more rapid reduction in the return concentration can be observed in LSSE #2 than in LSSE #1. Consequently, the duration of LSSE #2 with a detectable return concentration (also called the squeeze lifetime) lasted for only 20 PV until the return concentration dropped below 2 mg L^−1^ and another 228 PV to 0.5 mg L^−1^. On the other hand, the observed return concentration in LSSE #1 dropped to around 2 mg L^−1^ at 120 PV and stayed relatively stable at this level with minor fluctuations for another 720 PV. BaPCI material in LSSE #1 exhibited a substantially prolonged squeeze lifetime of 840 PV when the inhibitor return concentration was maintained above 2 mg L^−1^. From the viewpoint of mass balance, it was calculated that over 97% of the total amount of PSS injected had been returned by the end of LSSE #2, whereas this number was only 83% in LSSE #1 (Table 4). It can be postulated that if the return concentration post 840 PV in LSSE #1 can be maintained at the 2 mg L^−1^ level, the retained inhibitor can sustain another 450 PV of synthetic brine return, extending the total return volume to 1290 PV. The observed extended squeeze lifetime of BaPCI can be attributed to 1) high affixation to the surface of the calcite medium, as discussed above, and 2) a lower aqueous solubility in the synthetic brine solution. Extensive discussions can be found in the previous studies investigating the synthesis and return behavior of phosphonate-based scale inhibitor colloid/nanomaterials [17,21]. These authors argued that the aqueous solubility of the colloidal inhibitor materials dictates the inhibitor return behaviors in that the retained inhibitor materials on formation medium surfaces would gradually release from the medium surface controlled by the solubility of such materials in a brine solution. A low aqueous solubility will lead to a more gradual release of the inhibitor materials and hence an extended squeeze lifetime. To put into perspective of actual field scale control operations and potential economic impact, a normalized squeeze lifetime (NSL, m^3^ kg^−1^) has been calculated following the approach reported in previous studies [18] as: (2)$$NSL\;\left(m^{3}kg^{-1}\right)=\frac{return\;volume\;\left(L\right)\times 10^{6}\left(mg\;kg^{-1}\right)}{inhibitor\;mass\;\left(mg\right)\times 1000\;\left(L\;m^{-3}\right)}$$

NSL characterizes the potential volume of produced brine to be protected given the mass of 1 kg inhibitor delivered into the formation during the field inhibitor return. By plugging in the return volumes from LSSE #1 and #2, NSL values of 152 and 4 m^3^ kg^−1^ can be calculated, as shown in Table 4. This suggests that during field inhibitor return, a mass of BaPCI material with 1 kg of PSS embedded can control the scale threat for a volume of 152 m^3^ of produced water with a PSS return concentration of 2 mg L^−1^ or higher. Similarly, a scale protection duration (SPD) value can be calculated for both the BaPCI and PSS inhibitors following [21]: (3)$$SPD\;\left(d\right)=\frac{NSL\;\left(m^{3}kg^{-1}\right)\times DTPMP\;mass\;\left(kg\right)}{Production\;rate\;\left(m^{3}d^{-1}\right)}=\frac{NSL\;\left(m^{3}kg^{-1}\right)\times 100\;kg}{150\;m^{3}d^{-1}}$$

SPD concerns the duration of scale protection, assuming a production rate of 150 m^3^ d^−1^ given 100 kg of inhibitor delivered into the formation. The SPD values for LSSE #1 and #2 can be calculated as 101 and 2.7 d. In other words, the BaPCI material with 100 kg PSS embedded can yield a protection time of 101 days, considering a production rate of 150 m^3^ d^−1^. Compared with the PSS inhibitor, the significantly improved NSL and SPD values for BaPCI are attributed to the prolonged inhibitor return volume. With regard to future investigations, experimental studies can be carried out in several fields, such as investigating the feasibility of adopting other polymeric inhibitors with different functional groups and molecular mass for the preparation of scale inhibitor colloidal materials; conducting in-depth characterization of the polymers as well as the prepared scale inhibitor colloidal materials; systematically investigating the return behaviors of polymeric inhibitors and polymer-based scale inhibitor colloidal materials. 

## 4. Conclusions

In this study, a novel BaPCI material composed of the barium-PSS inhibitor has been prepared by a synthesis route involving precipitation and surfactant surface modification. The optimal synthesis condition was determined to be in the presence of 0.2% SDBS and 0.2% KCl with a sonication time of 10 min. The prepared BaPCI material possesses a particle size of 230 nm and zeta potential of −63 mV with a Ba to PSS monomer ratio of 1:2. Instrumental characterizations indicate that, during the synthesis process, Ba species interacted with the sulfonate group of PSS, forming Ba-sulfonate bond and that BaPCI material still carries a number of characteristic groups of the PSS monomer. Scale control tests suggest that the inhibition performance of BaPCI against common scales of calcite and calcium phosphate is on par with that of PSS given the same inhibitor concentration, and an increase in BaPCI concentration can lead to an enhanced scale inhibition performance. Laboratory transport studies in calcite-packed columns suggest that the migration capacity of the BaPCI can be enhanced with SDBS surfactant preflush. Moreover, an increase in temperature from 4 °C to 70 °C, an elevation in flow rate from 5.5 to 96 m d^−1^, and a reduction in calcite grain size can reduce the breakthrough efficiency of BaPCI. According to the advection-and-diffusion mechanism, the transportability of BaPCI in the calcite-packed column is mainly influenced by surface adsorption and deposition of BaPCI to the calcite medium. LSSEs indicate that BaPCI can return PSS inhibitor in a much more gradual manner with a significantly reduced inhibitor return mass in the early stage of LSSE compared with the PSS inhibitor. As a result, the squeeze lifetime of BaPCI is over 40 times longer than that of PSS, which can be characterized by the calculated NSL and SPD values. The extended squeeze lifetime and enhanced return performance of BaPCI are due to high affixation to the surface of the calcite medium and lower aqueous solubility in the brine solution. The prepared novel BaPCI material has a great potential to be adopted for field scale control where environmentally friendly, thermal stable, and/or calcium tolerating requirements should be satisfied. This study further expands and promotes our capacity to fabricate and utilize functional colloidal materials for mineral scale control. 

## Data Availability

The data presented in this study are available on request from the corresponding author.

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
