# Peer review of "Preparation and Laboratory Testing of Polymeric Scale Inhibitor Colloidal Materials for Oilfield Mineral Scale Control"

_polymers, 2022, doi:10.3390/polym14194240_

Round 1

Reviewer 2 Report

Journal: Polymers

Title: Synthesis and laboratory testing of polymeric scale inhibitor colloidal materials for oilfield mineral scale control

Author: Hanji Wang, Huaxia Dong, Xianbin Liu, Ping Zhang

Manuscript ID: polymers-1912128

This paper focuses on the synthesis of novel BaPCI material composed of barium-PSS inhibitor prepared by a synthesis route involving precipitation and surfactant surface modification.

I think it is an interesting and complete study where synthesis and characterization are perfectly developed. Abbreviations: are useful to follow the work. Introduction: authors perfectly describe the problem and the alternatives to avoid the mineral scales.

For this reason, my recommendation is to accept the paper for publication, nevertheless, the following points should be better explained or included:

What is the reason to use this molecular weight, has the authors check another molecular weight of polymer?

Experimental procedure is clearly described, but authors could explain the reason of temperature chosen.

Which is the temperature of transport experiment?

Characterization techniques are useful to obtain the results, but I think more polymer characterization should be included.

Line 379: oilfields operations could be named.

Which is the error of turbidity method of analysis?

Round 2

Reviewer 1 Report

I am satisfied by the correction done by the authors